# Habit—Does It Matter? Bringing Habit and Emotion into the Development of Consumer’s Food Waste Reduction Behavior with the Lens of the Theory of Interpersonal Behavior

**DOI:** 10.3390/ijerph19106312

**Published:** 2022-05-23

**Authors:** Sumia Mumtaz, Amanda M. Y. Chu, Saman Attiq, Hassan Jalil Shah, Wing-Keung Wong

**Affiliations:** 1Air University School of Management, Air University Islamabad, Islamabad 54000, Pakistan; sumiamumtaz@gmail.com (S.M.); saman.attiq@mail.au.edu.pk (S.A.); 2Department of Social Sciences, The Education University of Hong Kong, Tai Po, Hong Kong, China; 3NUST School of Social Sciences and Humanities, National University of Science and Technology, Islamabad 44000, Pakistan; hassan6ff@gmail.com; 4Department of Finance, Fintech & Blockchain Research Center, and Big Data Research Center, Asia University, Taichung City 41354, Taiwan; 5Department of Medical Research, China Medical University, Taichung City 40447, Taiwan; 6Department of Economics and Finance, The Hang Seng University of Hong Kong, Shatin, Hong Kong, China

**Keywords:** positive emotions, awareness of consequences, environmental knowledge, social norms, habits, facilitating conditions, waste reduction intentions

## Abstract

The immense food waste, generated by restaurants is not only a serious burden for the foodservice business but also a cause of anguish for the emerging nations in which eating out is becoming increasingly trendy. Consumers’ food wastes account for a significant portion of restaurant food waste, indicating the need for a change in consumers’ behavior to minimize food waste. To examine this problem, our study sought to identify the elements that influence restaurant consumers’ behaviors on food waste reduction, reuse, and recycling. The influence of anticipated positive emotions, awareness of consequences, environmental knowledge, and social norms on waste reduction intentions were examined by using a quantitative technique in the investigation. Furthermore, the influence of habits, waste reduction intentions, and facilitating conditions on food waste reduction, reuse, and recycling behaviors have also been investigated. The study collected 1063 responses and employed the PLS-SEM approach to verify the hypotheses. The results suggested that anticipated positive emotions, awareness of consequences, environmental knowledge, and social norms all have substantial impacts on waste reduction intentions. In addition, habits, waste reduction intentions, and facilitating conditions have noteworthy influences on consumers’ behaviors towards food waste reduction, reuse, and recycling in restaurants. Understanding these elements could help in correcting customers’ waste behaviors in restaurants. The findings in this study are useful for managers, policymakers, and researchers who want to solve the problems of food waste. The implications, limits, and suggestions for further studies have also been discussed in our study.

## 1. Introduction

Food waste is not only a worldwide problem and a societal challenge but also a source of causing food poverty [1]. Nearly a third of all edible food in the world is wasted (i.e., 1.3 billion tons/year), resulting in an annual financial loss of roughly USD 750 billion. Resultantly, a global debate has ensued to find ways of reducing food waste as well as developing a more sustainable society [2]. Food waste ranges from 280 to 300 kg/capita/year in developed countries, whereas, 120 to 170 kg/capita/year food is wasted in under-developed countries. This enormous wastage can be effectively reduced [3]. Goal 12 of UN-approved Sustainable Development Goals (SDG) relates to “reducing and recycling waste and choosing sustainable products”, goal 12.3 envisions to minimize per capita food waste by half by 2030. Food waste (FW), part of solid waste, is disposed-off or thrown away by consumers [4]. Consumer food waste in its edible and consumable condition is a major issue with economic (wastage of money), environmental, and societal ramifications, leading to phenomena such as inequality, food security, and famine [5]. Food waste also affects land and water, and it produces greenhouse gases [6]. Ironically, the social, environmental, and economic impacts notwithstanding, studies on consumer behavior towards FW are quite limited.

Pakistan’s rapid urbanization and population growth have resulted in the generation of approximately 87,000 tons of solid waste per day, which is increasing at a rate of 2.5 percent annually [7]. Pakistan creates 1.896 to 4.29 kg of MSW per capita on a daily basis. Pakistan ranks 11th on the Food Security Risk Index (roughly 61 million people are food insecure), making it one of the world’s most food-insecure countries [3]. Food waste collection and management are more complex and challenging, but reducing the food waste is the responsibility of all stakeholders (farmers, industry, merchants, media, and consumers) [8]. Regardless of the seriousness of this issue in Pakistan, the question of consumer food waste reduction has not received significant consideration. 

Whereas, sufficient research has been undertaken in Pakistan in the areas like solid waste and plastic waste management, research on consumer food waste reduction is yet to receive due attention [9].

### 1.1. Significance of the Study

The existing research on food waste remains far from satisfactory. To circumvent the limitations of the existing literature, in this paper, we propose to extend the theories developed in the literature by exploring three major perspectives (contextual, theoretical, and practical) to improve the significance of the theory, as stated in the following subsections.

#### 1.1.1. Contextual Significance

First, the estimated cost of FW in the developing world is between three hundred and ten billion US dollars. The food service business (restaurants and hotels) is largely associated with the overuse of energy, greenhouse gas emissions, and heavy waste of food. However, the consumer’s part in FW in this industry is quite significant with respect to their beliefs and past behavior [10]. According to one study, a large percentage of food is wasted at the last stage of the food distribution network, which is at the consumer level. Unfortunately, very few restaurants emphasize food waste reduction and recycling [11]. Consumer food waste behavior research examined the antecedents of behavior, but at restaurants, not much was found in detail. Second, consumer food waste behavior has economic, social, and environmental ramifications (hunger, lessening of natural and financial assets, ecosystem, etc.) [12]. While there has been a growing focus on consumer food waste, its complexity is still unclear, and food waste reduction is a significant issue as it is linked to consumer behavior [13]. Third, food waste reduction behaviors, such as recycling, that could reduce waste have been developed by an international study [14]. Variations exist in the appearance of waste-related behaviors such as reducing, reusing, and recycling [15]. Consumers can reduce, reuse, and recycle extra food products [16]. According to the study, food waste reduction should be the most essential part of waste reduction initiatives, with reuse and recycling being secondary priorities [17].

#### 1.1.2. Theoretical Significance

Scholars have proposed the idea of 3R (i.e., reduce, reuse, recycle), for defining and measuring food waste reduction behavior [17,18]. Research is needed in the areas of 3R that combines the three waste reduction behaviors of reducing, reusing, and recycling [19]. In this direction, food waste is more common downstream in the supply chain, especially, at the point of sale and at the point of consumption (consumer) [20]. Comparison has shown that the propensity of FW is more pronounced in younger consumers than mature consumers (over 65 years of age) [21]. Second, most studies on consumer behavior and food waste have focused on household consumers [22]. However, the food service sector has recently received some attention, with a focus on the quantity of waste rather than consumer behavior to reduce waste [23]. Third, food waste behavior has been widely researched using the notion of planned behavior. Recent research highlight that measuring consumer behavior by combining TPB with other theories will provide more significant results and suggest the integration of different theoretical lenses to comprehensively measure FWRB, such as the TIB, affect, social practice theory, and environmental psychology [24,25]. TPB is the most common model used to explain behavior through intention, but it lacks feelings and habits and lacks the power to explain emotional and habitual behavior [26]. Fifth, the research findings of the existing literature reveals inconsistent relationships between variables such as awareness of the consequences and environmental knowledge with waste reduction behaviors [27]. Furthermore, it is thought that a person’s chance of engaging in behavior is linked to his or her habits, as well as the conditions that facilitate the behavior and intention. Hence, it is suggested that theories that ignore the role of habit and emotions in their models make it impossible to accurately assess the psychological antecedents of FW behavior [24]. The present investigation attempts to fill the above-mentioned voids in the research by establishing a conceptual framework related to interpersonal behavior theory and examining the determinants that influence waste reduction behaviors such as reducing, reusing, and recycling information from young consumers in the setting of restaurants. This research helps to comprehend the consumer’s point of view from both a management and theoretical one. The research would, in theory, evaluate the factors quantitatively and look at their direct influence on young consumers’ FW reduction intentions. Moreover, 3R behaviors will be examined through habits, waste reduction intentions, and facilitating conditions.

#### 1.1.3. Practical Significance

Our study will also help policymakers build the awareness of young consumers related to food waste. Young consumers might be made aware of the importance of participating in sustainable FWRB by adhering to rules and measures set in this respect. Therefore, initiatives and marketing that emphasize social influence have a greater possibility of encouraging consumers to help minimize food waste. Consumers’ behaviors about food waste should be changed by educating them on the benefits of reducing food waste. Practitioners should motivate young consumers to engage in FWRB by stressing food security and imprinting the consequences of FW on their minds. The younger generation should be taught about food waste reduction via social media and kept informed about the consequences. In its utility for the management, this research will benefit the food service business, such as restaurants and hotels, as well as the government, in developing plans to inspire FWR by reducing, reusing, and recycling food rather than wasting it.

The paper is organized as following: The first portion of the study outlines the research problem and highlights gaps in the literature. The literature review on variables is included in the second section. The method, analytical techniques, and results, as well as the study’s implications, limits, and recommendations, are all presented in the last portion of our paper.

## 2. Literature Review and Theoretical Foundation

Food waste is a growing concern worldwide, hence, including consumers in waste reduction strategies through their behavior is critical. Because of the complexities of human behavior, the most popular TPB appears inadequate to describe the emotional side of consumer behavior. The theory of planned behavior (TPB) is a deterministic theory that illustrates behavior by intention while not including emotional and non-conscious variables [28]. A theory of interpersonal behavior (TIB) has been employed to assist our research in order to overcome these problems. In contrast to the notion of planned behavior, TIB was introduced by [24]. It reveals the significance of habits and emotions in shaping intentions to involve in specific behaviors. Additionally, it claims that “intention is an outcome of effect, cognition of consequences, and social and personal norms”. Furthermore, it is suggested that the chance of carrying out behavior is related not just to the individual’s habits but also to conditions that facilitate the action and intentions. Since habit and emotions were not considered in the earlier research, they were unable to accurately quantify the psychological causes of consumer behavior regarding food waste [24]. The environment in emerging nations is being harmed by an absence of consumer food waste awareness of consequences and environmental knowledge [21]. Investigating the consequences of food waste is critical because it involves a change in consumer attitudes and behavior [20].

### 2.1. Consumer Behavior Reuse, Reduce, and Recycle (3R)

The key to reducing waste is in understanding what motivates consumers to reduce waste [27]. Behavior to reduce waste through recycling is recognized as an international study topic [14], and researchers have characterized waste-related behaviors such as reducing, reusing, and recycling [22] as ways that consumers can reduce, reuse, and recycle food products [16]. The involvement of consumers in food waste is critical; hence, a complete grasp of the elements that influence customers’ attitudes and behavior around food waste is required [29]. Consumer FWR in the restaurant business is linked to reducing and minimizing food waste by practicing reducing it, using recycling practices, and sorting waste [30].

Consumer food waste behavior takes place at all stages, from planning to consumption. Reducing food waste behavior entails lowering the food supply and increasing consumption by consuming what has been purchased. Food waste reuse behavior is based on the reuse of food items, such as using or eating them again in the future or sharing them with others. Food waste recycling behavior includes separating food from other types of waste. Consequently, consumers who undertake these three behaviors (reduce, reuse, and recycling) will be more pro-environmental and will also reduce food waste [17]. Researchers have focused on one type of waste behavior at a time for investigation, such as recycling, reusing, or minimizing waste, but only a few studies have considered the 3R behaviors of waste reduction collectively [19]. Our investigation focuses on the 3Rs to evaluate FW behavior (reduce, reuse, and recycle) and how it may be reduced, reused, or recycled to be utilized as a raw resource in composites.

### 2.2. Anticipated Positive Emotions (APE)

Anticipated emotions are described as “one’s expected emotional responses that one will experience by engaging in a particular behavior in the future,” such as the anticipated positive emotions (APE) of pride, confidence, as well as accomplishment [29]. When consumers have more positive emotions about a hotel, they are more likely to use it and recommend it to others, or vice versa. APE has a direct influence on intention [31], and it has a considerable detrimental influence on the intention to conserve power [32]. APE has an insignificant influence on the intention to consume recycled water [33]. The positive effect of anticipating emotion of pride was found to be a highly influential factor that changed the environment and consumer FWRB in restaurants [34]. APE of pride has established a positive influence on young consumers’ intention to reduce waste at tourist destinations [35]. APE influences consumer waste reduction behavior and saves water while staying at a hotel when a hotel’s waste reduction practices evoke positive feelings in consumers and encourage them to both cut down on waste and conserve water. According to another study, consumers’ behavioral intentions are unaffected by anticipated positive emotions regarding food waste reduction [24]. The APE of pride influences green behavior significantly [36]. Thus, the hypothesis is

**Hypothesis** **(H1):**Anticipated positive emotions have a significant impact on young consumers’ food waste reduction intentions.

### 2.3. Social Norms (SN)

Subjective norms are a consumers’ ideas about whether it is acceptable to behave in a specific way in the face of societal pressure [37], through which the views of a reference group or system, such as family, classmates, friends, and the community in general, influence a person’s beliefs, attitudes, feelings, and judgments. SN is termed as “rules and standards that are understood by members of a group and that guide and/or constrain human behavior without the force of laws”. Subjective norms are typically formed as a result of peer pressure, friends, or family members, forcing individuals to adhere to specific pre-specified norms. Despite the fact that they showed a positive relationship, subjective norms were demonstrated to have an analytically negligible impact on recycling intentions. Surprisingly, the findings indicated no substantial influence of subjective norms on young consumers’ intentions to recycle e-waste [38]. Individuals are more inclined to engage in the behavior as long as they observe a higher subjective standard. Individuals are frequently encouraged to conform to the viewpoints or expectations of key referents. The link between subjective norm and intention was negative, indicating that the consumer’s intention to conserve energy is unaffected by important individuals (e.g., family members, friends, or neighbors) [32]. Subjective norms are interpreted as the extent to which other relevant people would accept or reject the person’s wasteful behavior. The results indicate that subjective norms demonstrate to be crucial determinants of food waste reduction intentions [22].

People take the influence of people they regard as significant as confirmation of agreement with the views of those other people. It is thought that customers usually adhere to social norms not just as a result of social pressure but also for other reasons because they provide information on what type of behavior is most suited [39]. The influence of social norms on waste reduction and recycling intentions was confirmed in a study, and they are important motivators of young tourists’ behavioral intentions [40]. Subjective norms influence certain behaviors, such as food waste, and have a significant effect on waste reduction intentions [13]. Therefore, it can be hypothesized as

**Hypothesis** **(H2):**Social norms have a significant impact on young consumers’ food waste reduction intentions.

### 2.4. Environmental Knowledge (EK)

Environmental knowledge (EK) relates to “a person’s knowledge about the environment” and reveals positive relations with intention and behavior. Studies have not found environmental knowledge to be solely responsible for pro-environmental behavior. Environmental knowledge has a substantial impact on the sustainable behaviors of rice growers. Researchers have found that highly knowledgeable consumers were more concerned about the environment and showed highly pro-environmental behavior both at home and at restaurants. knowledgeable consumers preferred to support those restaurants that practice food waste reduction activities, such as reducing, reusing, recycling, etc. [41]. Environmental knowledge is related to issues and solutions, which in turn influence the decision-making of consumers and enhance the tendency towards green behavior. Consumers tend to purchase eco-friendly things as a result of the direct effect of environmental knowledge on food waste concerns. Results of another study revealed that students who have EK and education are more prone to behave in favor of the environment [42] associated with energy conservation behavior [43]. They also found a significant impact of EK on recycling intentions at the consumer level but not at the organizational level [44]. Therefore, the EK-related hypothesis is that

**Hypothesis** **(H3):**Environmental knowledge has a significant impact on young consumers’ food waste reduction intentions.

### 2.5. Awareness of Consequences (AC)

“Awareness of consequences (AC)” is described as “the cognition that an individual believes that failure to perform a specific behavior may bring adverse consequences to others” [45]. In simple words, AC explains the extent to which one is aware of the repercussions if environmental information is communicated. The awareness of the consequences impacts the attitude that her or his activities may have an impact on others if she/he does not modify her/his behavior [46]. As people become more aware of the consequences, they induce a more positive evaluation of sharing behavior, with a stronger social pressure and moral obligation to behave. It denotes a consumer’s awareness of whether or not to conduct a behavior, such as tourists’ awareness of consequences, which influences their intention to demonstrate pro-environmental behavior at national parks [47]. Similarly, when individuals are aware of the environmental consequences of recycling, they chose to put their energies into recycling their waste, and they discovered a positive, significant connection with the intention of recycling [18]. The greater the students’ awareness of environmental consequences, the greater their behavioral intention demonstrated reuse behavior toward recycled water, which had an indirect significant positive impact on their intention to reduce when aware of environmental consequences such as noise, traffic congestion, greenhouse gases, and global warming [48] and displayed waste sorting behavior (recycling) [45]. One study also found that AC did not have a substantial effect on usage intention, while they understood the sustainable consequences [49]. Although AC has been shown to determine recycling behavior intention, the role of society, peers, and groups in supporting recycling is important, as it influences consumers to recycle as well [29]. Similarly, awareness, along with social norms and behavioral control, is an important factor for food waste reduction intention, indicating that consumer FWRB in the restaurant business is highly influential and changes the environment [34]. It was discovered that when customers are aware of the environmental and consumer repercussions of food waste, they opt to reuse, donate, or resell their leftover food. The study found a positive influence on reuse behavior and on donation behavior but not on reselling behavior [50]. Therefore, the hypothesis about AC in the study is

**Hypothesis** **(H4):**Awareness of consequences has a significant impact on young consumers’ food waste reduction intentions.

### 2.6. Waste Reduction Intentions and Waste Reduction Behaviors (Reduce, Reuse, Recycle)

Generally intention is defined as “the likelihood of an individual performing the behavior in the future” [37]. Behavioral intention is described as an inclination to act or not to act on a specific task, whereas an individual’s intention to execute a behavior causes his or her recycling behavior to change positively and significantly [51]. The association of FW reduction intention with the amount of FW was proved to be negative, suggesting that FW reduction intention assists in decreasing the behavior towards FW reduction [12]. According to the findings, the study respondents’ FW reduction intention has a negative impact on their FWB. Additionally, those with a stronger FW reduction intention generated less waste [22]. Furthermore, food waste reduction intention was a substantial determinant of restaurant FWRB; the higher the intention, the less likely the food waste reduction behavior [52]. Behavioral intention strongly predicted reuse behavior, indicating that people were more likely to reuse products rather than discard them after only using them once [18]. Therefore, it can be hypothesized that:

**Hypothesis** **(H5):**Waste reduction intention has a significant impact on young consumers’ food waste reduction behavior (reduce food waste).

**Hypothesis** **(H6):**Waste reduction intention has a significant impact on young consumers’ food waste reduction behavior (reuse food waste). 

**Hypothesis** **(H7):**Waste reduction intention has a significant impact on young consumers’ food waste reduction behavior (recycle food waste). 

### 2.7. Facilitating Conditions (FC)

Researchers have defined “facilitating conditions” broadly as factors that “permit a person to behave” [53]. According to the theory of interpersonal behavior, habits, facilitating conditions, and behavioral intention, all impact behavior. Facilitating conditions are described as “the beliefs about the availability of resources to facilitate behavior” [54]. Facilitating conditions as a variable were used as an antecedent of the 3R behaviors (reduce, reuse, and recycle), and it can be explained in this manner that facilitating conditions are what people think about the resources and support they have available to do a certain thing. ‘Facilitating conditions’ involves contextual or situational prospects for being involved in a specific behavior as well as limitations that inhibit it from being adopted [55]. FC influences real behavior that facilitates the occurrence of certain behaviors. Furthermore, FC moderates the intention–behavior relation [28]. Facilitating conditions is a situational factor or prospect that persuades or dissuades an individual related to a particular behavior, and evidence proved that FC had a significant and positive influence on 3Rs behaviors (reduce, reuse, and recycle) [56]. Researchers found FC had a weak relationship to 3Rs behaviors (reduce, reuse, and recycle), indicating that the lack of facilities influences the behavior and that if necessary facilities are available, then consumers will perform 3Rs behaviors [57]. Therefore, it can be hypothesized as

**Hypothesis** **(H8):**Facilitating conditions have a significant impact on young consumers’ food waste reduction behavior (reduce food waste).

**Hypothesis** **(H9):**Facilitating conditions have a significant impact on young consumers’ food waste reduction behavior (reuse food waste). 

**Hypothesis** **(H10):**Facilitating conditions have a significant impact on young consumers’ food waste reduction behavior (recycle food waste). 

### 2.8. Habit (HB)

Habits are termed as a “learned automatic response that maintains repetitive actions in certain situations” [58]. Basically, it is a psychological aspect that predicts behavior. The primary component measured to impact e-waste recycling behavior was habit [59], greatly affecting intention towards e-waste [38]. Other environmentally friendly behaviors include reusing and reducing waste by feeding food scraps to animals [60]. The findings support a broader paradigm of food waste behavior. In fact, this corresponds to interpersonal behavior theory, and the results confirmed that food waste behavior is significantly predicted by both emotions and habits [24]. 

Researchers observed that there was a favorable effect on waste reduction intention as well as behavior [29] and were proved to be a substantial antecedent of FWRB [58,61,62]. The 3Rs behavior shows a substantial positive association with habit. However, individuals with no apparent recycling habit had a greater previous behavior–intention connection, according to the data. As a habit, it showed the greatest significant connection with 3Rs behavior [57]. As a result, derived from the previous research, it is possible to propose:

**Hypothesis** **(H11):**Habit has a significant impact on young consumers’ food waste reduction behavior.

**Hypothesis** **(H12):**Habit has a significant impact on young consumers’ food waste reusing behavior.

**Hypothesis** **(H13):**Habit has a significant impact on young consumers’ food waste recycling behavior.

### 2.9. Mediation of Waste Reduction Intentions 

Although emotion established a larger impact on WRI, the findings also showed that it has little impact on activating and modifying behavior via the role of intention as a mediating factor, indicating that a number of participants are emotionally concerned regarding environmental issues caused by food waste and increased hunger as a result of food waste patterns [63]. The findings indicated that behavioral intention mediated the impact of EK and awareness on behavior. According to the findings, consumer awareness and social norms have a major impact on behavior, which is mediated by the consumer’s intention to minimize or reduce food waste [64]. Although emotion had the greatest impact on the waste reduction intention, the findings also demonstrated that it has little impact on activating and modifying behavior through the role of intention as a mediating factor, indicating that many respondents are emotionally concerned about environmental issues caused by food waste and raise hunger as a result of food waste patterns [63]. It implies that independent of their intent to minimize or reduce food waste, awareness and social norms may have an effect on young customers’ buying behavior. Subjective norms and marketing/sales addiction, though, have no substantial impact on food waste reduction through the mediation of WRI [65]. The findings revealed that consumer attitude and social norms are strongly mediated by the influence of behavioral intention on food waste behavior, while social norms had a large influence on food waste behavior, and between both factors, the behavioral intention had a significant mediating effect as well [23]. Food waste behavior is negatively influenced by the food WRI, while intention mediates between attitude, subjective norms, PBC, and FW behavior [22]. Return/recycling intentions mediate the influence of attitude, PBC, subjective norms, moral norms, awareness of consequences, convenience, and waste recycling behavior, suggesting that consumers prefer to reuse rather than throw away their items. Therefore, it can be hypothesized that:

**Hypothesis** **(H14):**Waste reduction intention mediates the impact of anticipated positive emotions on waste reduction behavior.

**Hypothesis** **(H15):**Waste reduction intention mediates the impact of social norms on waste reduction behavior

**Hypothesis** **(H16):**Waste reduction intention mediates the impact of environmental knowledge on waste reduction behavior.

**Hypothesis** **(H17):**Waste reduction intention mediates the impact of awareness of consequences on waste reduction behavior.

The figure below represents the conceptual framework of the study. The arrows indicate all hypotheses (Figure 1).

## 3. Methodology

### 3.1. Research Design and Measurement Scales

In this study, we apply both a quantitative research method and descriptive research design to analyze the study objectives. The purpose of conducting quantitative analysis in the study is to determine the relationship between all variables based on the theory of interpersonal behavior (TIB). To obtain information from the intended research participants, we used a structured questionnaire based on a survey based on deductive research approach. On the grounds of predetermined scales, all the variables used in our study have been adapted and conceptualized. Using data from the first section of the questionnaire, the respondents’ demographic information was analyzed; whereas, in the next section of the questionnaire, the variables used in the research were measured. The constructed measurement items consisted of items from various references, and a 5-point Likert scale was used from 1 (strongly disagree) to 5 (strongly agree) to obtain replies. A group of professionals (3 research experts and 1 professional) with experience in food waste investigation evaluated the validity of the created survey tools. 

Based on the input from experts, the measuring points were also revised for clarity. The items in the questionnaire, adopted from previous well-known research, were written clearly and expressed in simple sentences so that participants could understand all the questions clearly. Items assessing anticipated positive emotions were adapted from [41], with the instrument in the study having four questions evaluating anticipated positive emotions. [18] designed a social norms scale with five items that were adapted for this study. It also used a 5-item instrument to assess the concept of awareness of consequences from [18] whereas, six items of habit scale is used by [66]. Seven items, adapted from [67,68] are used to quantify environmental knowledge. Three waste reduction intention items, established by [63], were adopted. Three items from the instrument in [69] were utilized to quantify facilitating conditions. Reference [70] developed the scales of “reduce, reuse, and recycle behaviors”, and five items for each behavior were adopted from their instruments.

### 3.2. Data Collection

We adopted a cross-sectional survey approach, employed the approach of an online survey, and requested Prolific pick volunteers residing in the twin cities of Islamabad and Rawalpindi in Pakistan to obtain data from Pakistani universities. The participants were asked to take part in the survey willingly. Participants were assured of the confidentiality and anonymity of their information prior to undertaking any reply. The survey questionnaire consists of two sections: demographic questions were asked in the first part, and questions were asked to measure the variables being studied in our paper in the second part. A screening question was asked at the start of the survey regarding the food waste, “do you leave food after meal”? Using a convenience sampling technique, the responses of a total of 1063 university students from the twin cities, (Islamabad and Rawalpindi), were solicited to obviate the chances of a small sample size. The received completed questionnaires were cross-checked to exclude partial replies, resulting in a total of 1063 legitimate responses (fully completed, no missing data) to be used in the statistical test.

### 3.3. Method of Data Analysis

The partial least squares structural equation modeling (PLS-SEM) approach, which is better than many traditional approaches is a method for predictive analysis that optimizes the variance explained by the latent endogenous elements. Thus, we employed the PLS equation modeling in our study to examine the postulated model. This technique is one of the best approaches being used to examine complicated relationships between components or numerous constructs [71]. The PLS-SEM analysis covers indicator reliability, internal consistency reliability, convergent validity, discriminant validity, average variance extracted, effect size, path coefficient estimations, and predictive relevance [72]. This method is also useful for testing hypotheses. It can also be used to investigate the causation link between latent components. The PLS-SEM is a 2-step approach in which the first step is known as an “inner model,”, whereas the second stage is known as an “outer model”. Both stages use distinct ways to validate the research model [73].

## 4. Results and Analysis

### 4.1. Descriptive Analysis

The demographic characteristics of the sample, consisting of the gender, age, and education of the 1063 people, were taken into account, corresponding to 572 males and 491 females. There were 449 people between 18 and 25 years of age. The number of participants between 26 and 35 years of age was 313, 104 were between 36 and 45 years of age, and 68 had more than 45 years of age. The majority of respondents (522) hold a bachelor’s degree. The results are illustrated in Table 1.

### 4.2. Measurement Model

The PLS-SEM modeling approach was applied in this work. In the inner model, the outer loads were first assessed. Verifying the reported variables and their linked items improves the inner model’s validity. The outer loadings of each item in all constructs were examined to validate them. However, none of the items have an outer loading of less than 0.50. Therefore, the study keeps all the items in the instrument. Four items were used to assess anticipated positive emotions (i.e., APE1, APE2, APE3, and APE4). Five items were used to assess social norms (i.e., SNM1, SNM2, SNM3, SNM4, and SNM5). Five items were used to assess awareness of consequences (i.e., AOC1, AOC2, AOC3, AOC4, and AOC5), while environmental knowledge was evaluated using seven elements (i.e., EKW1, EKW2, EKW3, EKE4, EKW5, EKW6, and EKW7). Six items were used to assess habit (HAB1, HAB2, HAB3, HAB4, HAB5, and HAB6), with no item being excluded. 

The facilitating condition was assessed using three items (FCR1, FCR2, and FCR3), and there was no item removed. Three items were used to measure waste reduction intention (WRI1, WRI2, and WRI3). Five items were used to evaluate reduction behavior (i.e., RED1, RED2, RED3, RED4, and RED5), and reusing behavior (i.e., REU1, REU2, REU3, REU4, and REU5), with no item being omitted; as well as recycling behavior of food waste (REC1, REC2, REC3, REC4, and REC5), with no item being omitted. Table 2 displays the outer loading findings.

#### Reliability and Validity Analysis

The two basic criteria for analyzing reliability are composite reliability and Cronbach’s alpha (internal consistency). Cronbach’s alpha was the first criterion employed. If the level of reliability in all parameters remains constant, it offers the reliability evaluation considering correlation among variables. The results of the Cronbach alpha for all variables ranged from 0.775 to 0.936, indicating a greater degree of reliability [74], i.e., >0.70. The outcomes are shown in Table 2. The results of Cronbach’s alpha indicated that all constructs have values greater than 0.70, which is an indication of reliability. 

The second indicator for internal consistency is composite reliability. The composite reliability threshold value is 0.7 [71]. As indicated in Table 2, the results of all the composite reliabilities for all constructs have a greater than 0.86, suggesting the reliability of all constructs.

Furthermore, the Dillon–Goldstein rho estimates of all constructs exceeded 0.8, suggesting the reliability of all of the constructs. This also validated the items’ reliability [71,72]. All variables’ outer loadings were used to assess the composite reliability. According to [72], the final composite reliability estimates ranged from 0.868 to 0.945, indicating good reliability (i.e., greater than 0.70). Table 2 displays the findings. Convergent validity is employed to examine the relationship among all elements in a variable. The average extracted variance (AVE) is used to examine the convergent validity of variables. According to researchers, the AVE value should be more than 0.50 to establish convergent validity [71,72]. The estimates of AVE of all constructs were greater than 0.50, demonstrating adequate convergent validity. Variables scored between 0.59 and 0.71 for convergent validity, according to the findings. Because AVE is greater than the required threshold of 0.50, all constructs indicate good convergent validity [75]. 

The Fornell–Larcker criteria were employed to determine discriminant validity, and the estimates of AVE of all indicators ought to be larger than the greatest squared correlation of the construct with another construct [71,72]. As seen in Table 3, all constructions were able to fulfill this condition.

### 4.3. Structural Model

After the assessment of the inner model, the outer model was used to test research hypotheses among all components. Five metrics were proposed by [76] for assessing the outer models: multi-collinearity assessment, hypotheses testing, evaluation of R^2^, assessment of effect size f^2^, and assessment of predictive significance Q^2^. In the following phase, the hypotheses are tested. The first hypothesis, H_1_, stated that anticipated positive emotions had a significant positive consequence on waste reduction intention (β = 0.187; t = 4.576). The second hypothesis, H_2_, states that social norms make a significant and positive effect on waste reduction intention (β = 0.1710; t = 4.345). The third hypothesis H_3_ states that environmental knowledge influences waste reduction intention positively (β = 0.224; t = 6.600). H_4_ was the fourth hypothesis, and it stated that awareness of consequences had a direct and significant effect on waste reduction intention (β = 0.315; t = 7.095). 

Waste reduction intentions showed a significant influence on waste reduction behavior, waste reuse behavior, and waste recycling behavior, according to the fifth H_5_ (β = 0.257; t = 8.715), sixth H_6_ (β = 0.510; t = 18.827), and seventh H_7_ hypotheses (β = 0.162; t = 6.147). H_8_ (β = 0.449; t = 14.948), H_9_ (β = 0.229; t = 8.337), and H_10_ (β = 0.145; t = 5.343) hypothesized that facilitating conditions influenced waste reduction, reuse, and recycling behavior. According to hypotheses H_11_ (β = 0.244; t = 8.440), H_12_ (β = 0.230; t = 8.503), and H_13_ (β = 0.650; t = 26.558), habit has a significant influence on waste reduction, waste reuse, and waste recycling behaviors. All significant hypotheses with *p* less than 0.00 were accepted and are shown in Table 4.

After checking the individual hypotheses, the prediction accuracy of the model was then measured using R^2^ (determination coefficient). As shown in Table 5, the coefficient of determination (R^2^) of all dependent latent constructs was deemed moderate to strong and thus acceptable.

Next, Table 5 displays the Q^2^ estimates, which examines the predictive significance of all constructs. The Q^2^ results examine the comparative predictive importance of an independent variable on a dependent variable, and an estimate greater than 0 suggests the acceptability of the correctness of the model’s path [72]. Table 5 presented the results of Q^2^, which were more than zero, demonstrating the predictive relevance of the constructs (i.e., anticipated positive emotion, social norms, environmental knowledge, and awareness of consequences) on waste reduction intention (0.519) and the predictive relevance that habits, facilitating conditions, and waste reduction intention have on the behaviors of young consumers who reduce, reuse, and recycle food waste. 

According to the findings, the Q^2^ estimates for food waste behaviors (reducing, reusing, and recycling) are 0.514, 0.521, and 0.496, respectively. That is, all estimates are over the threshold, i.e., more than zero [76]. All results can be analyzed in Figure 2 of structural model).

Furthermore, the effect size (f^2^) is calculated, and Hair et al. [76] suggested that f^2^ = 0.35 signifies a strong effect, f^2^ = 0.15 signifies a moderate effect, and f^2^ = 0.02 signifies a weak effect between the variables. The findings in Table 6 revealed that AOC has an f^2^ of 0.061 when it comes to food waste reduction intentions, indicating a moderate effect size (i.e., a value between 0.02 and 0.15). Anticipated positive emotions have an f^2^ of 0.033 when paired with food WRI, indicating a moderate impact size (i.e., a value between 0.02 and 0.15). Environmental knowledge has an f^2^ value of 0.054 when paired with food waste reduction intentions, indicating a moderate impact size (i.e., a value between 0.02 and 0.15). Social norms have a weak effect size. Facilitating conditions have a moderate effect size with recycle and reuse but strong with the reduce behavior. Habits have a strong effect size with recycle behavior, and moderate effect size with reduce and reuse behaviors. Waste reduction intention has a moderate effect size with recycle and reduce and strong effect size with reuse food waste behaviors.

### 4.4. Mediating Effects

Table 7 depicts the importance of waste reduction intention in mediating the relationship between anticipated positive emotions, social norms, environmental knowledge, and awareness of consequences and waste reduction behaviors (i.e., reduce, reuse, and recycle). Anticipated positive emotions (H_14_), social norms (H_15_), environmental knowledge (H_16_), and awareness of consequences (H_17_) all showed a substantial (*p*-values 0.00) indirect influence on waste reduction behaviors through the mediation of behavioral intention.

## 5. Discussion

The most effective strategy to reduce FW are to reduce, reuse, and recycle (3Rs). Food waste is being discouraged with all the efforts made to reduce food wastage across the world. Even though a declining trend has been observed in food waste, consumers’ behavior toward food waste continues to be a major issue with negative consequences for society. The intent of consumers to reduce food waste is influenced by anticipated positive emotions, social norms, environmental knowledge, and awareness of consequences. The FWRB of reducing, reusing, and recycling are directly influenced by habit, waste reduction intention, and facilitating conditions.

According to the findings in our analysis, anticipated positive emotions have direct positive impacts on waste reduction intention and indirect positive impacts on waste reduction behavior. As a result, consumers had positive emotions of pride in response to events involving food waste behavior, which is consistent with earlier studies. When consumers realize that wasting food will have negative impacts on their health [32], they are more likely to engage in appropriate behavior. Nonetheless, consumers, on the other hand, continue to waste food on a regular basis. Food waste is a serious problem in Pakistan because a huge portion of the population lacks access to sufficient food. Consumers are unaware that they are wasting food by leaving food on their plates, resulting in a financial loss. Consumers do not reuse their unconsumed food since they are not aware of the food waste consequences [18]. As a result, it is critical to disseminate knowledge about how food waste contributes to environmental, economic, and social issues [77]. A link was observed between positive emotions and recycling efforts, indicating that positive emotions reduce the negative sensations induced by wasteful behavior [78]. Anticipated positive emotions are an effective technique for persuading consumers to willingly reduce food waste in the food service business. As a result, individuals have a sense of pride and are more likely to be involved in the reduction of food waste than those who do not participate in such waste-reduction activities [79].

According to the findings of H2, social norms (SN) had a substantial impact on waste reduction intentions, indicating that the presence of positive responses offered by consumers as a consequence of the influence of other people accounted for the favorable impact of social norms on food waste reduction behavior. Social norms reveal that behavioral intention encourages practices such as FW minimization. However, the relevance of social norms dictates their effect, especially when socially negative acts become pervasive in a community. Some consumers’ behavioral intentions are motivated by norms as a result of factors that impact the way they interact with others, such as social norms [38]. The findings demonstrated that social norms had a minimal effect on intention; nonetheless, Pakistan has strong collectivist values, which would inspire young consumers with high levels of devotion and cohesiveness within their social groupings to follow norms. Young consumers place a high value on their friends, family, and social media before making any decisions, whether short-term or long-term [80]. In a comparable study [22], SN was shown to be a key indicator of food WRI. According to the study, customers’ intentions to decrease food waste in restaurants were not boosted by SN, contrary to previous research indicating no influence of SN on food waste reduction intentions. Because there is no societal pressure to do so, SN was unable to persuade consumers to take action to decrease restaurant food waste. Because of their shyness and the fact that food is wasted at restaurants, they may refrain from requesting that leftovers be wrapped and taken home [52].

The current study’s findings regarding H3 verified the influence of environmental knowledge (EK) on young consumers’ intentions to reduce waste, and the outcome was determined to be both positive and significant. The results are consistent with previous research [81]. In addition, the results also contradict previous research [82]. Furthermore, improving knowledge is crucial in order for individuals to understand their roles and obligations when it comes to decreasing food waste. Higher EK levels were found to have a favorable relationship with behavioral intent to prevent food waste. People who score better on the components that compose the knowledge construct are more concerned about environmental issues and use that knowledge as consumers plan to limit food waste. Nonetheless, this is in accordance with previous studies [43,83].

According to H4, the awareness of consequences (AOC) is a key aspect that supports reducing, reusing, and recycling FW through intent. As consumers grow more conscious of the negative effects of food waste on the environment and society, they choose to limit food waste and take satisfaction and pride in doing so. The findings confirm the theory that consumers’ awareness of the consequences of food waste builds attitudes that lead to socially acceptable activities to lessen the effects. Raising knowledge and awareness of the consequences of food waste is a key component of behavioral change. According to the findings, consumers are more involved in FW reduction when they are aware of the consequences of food waste reduction. The findings are consistent with the benefits of reducing food waste, which include reducing economic difficulties, environmental damage, and social factors such as hunger reduction. Our findings support the notion that increased consumer intention reduces the negative influence of food waste. Reference [84] discovered that reducing food waste has a larger contribution to socially responsible behaviors such as reducing, reusing, and recycling. Our study found that an absence of awareness among young consumers regarding the consequences of food waste, as well as a lack of awareness that wasting food is related to the wasting of resources such as money and energy, is a significant aspect that needs more investigation. In the industrialized world, consumers play an active role in recycling and reducing food waste. However, in Pakistan, food waste reduction, reuse, and recycling are not regarded as problems by consumers. Consumers who are aware of the environmental and social repercussions of food waste intend to reduce it. On the other hand, consumers keep wasting food on a regular basis. In Pakistan, the matter of FW is serious because a large number of people do not have sufficient food to eat. Consumers are unaware of the economic burden of wasting food and leaving food on their plates. Consumers avoid reusing leftover food because of their ignorance of the consequences of FW [18]. The study findings related to the hypothesis are therefore consistent with previous research. Reference [85] indicate that consumers with greater food WRI are concerned about the negative consequences of FW and participate in effective FWRB. Moreover, they consider that reducing FW is an important step toward lowering greenhouse gases and waste, as well as vital for preserving natural resources and making the environment healthy for forthcoming generations.

Based on the results of H5, this investigation found that WRI has a positive, significant direct impact on the behaviors of FW reduction, reuse, and recycling. This outcome is similar to earlier research that intention was impacting the behavior [18,51,57,63,65,86,87]. The findings demonstrate that intention is a substantial predictor of behavior and plays a critical role in modulating behavior. This implies that people are more inclined to decrease their waste by reusing it rather than discarding it after only one usage. Consumers’ confidence in their capacity to demonstrate their commitment to waste reduction was matched by their strong intention. As a result, to drive consumer waste reduction behaviors, it is vital to think about how to enhance consumers’ intention-to-behavior transformation process. Those with a higher intention of reducing food waste had a lesser amount of waste. Waste reduction intention considerably determines the behaviors of consumers in restaurants. The higher the intention, the less food is wasted in restaurants [52].

Respondents’ waste reduction behaviors are influenced by facilitating conditions, indicating that respondents’ waste reduction behaviors will improve if they have good facilities to reduce waste. Results are consistent with past studies [88]. However, the results contradict prior studies [66]. The existing FC for utilization is inadequate and is not always reflected in actual behavior. The data found that instead of reusing food, customers reduced and recycled it.

Furthermore, the study’s findings highlighted the contribution of habit in promoting FWRB, which is consistent with earlier studies (reduction, reuse, and recycling). The importance of habits in food waste reduction, reuse, and recycling behaviors cannot be overstated. Habit plays a critical role in determining whether we develop ecologically friendly behavior [89]. FWRB practices such as reducing, reusing, and recycling are regarded as pro-environmental, and if consumers establish the habit of reducing FW, they have a higher probability of embracing FW reduction wherever they eat at home, at restaurants, or elsewhere. Furthermore, the habit of reducing FW offers significant economic advantages. In underdeveloped nations such as Pakistan, habits might save cash and reduce FW. Although, reuse and reducing behaviors are influenced by habits [60], as are recycling behaviors [59]. However, this contradicts prior research [30]. According to the present study, habit seems to have an important and positive influence on young consumers’ reducing, reusing, and recycling behaviors, and similar results were found in previous research as well [57].

## 6. Conclusions

According to the findings of the research, all constructs, including anticipated positive emotions, social norms, environmental knowledge, awareness of consequences, waste reduction intentions, habits, and facilitating conditions, had a substantial impact on FW reduction, reuse, and recycling behavior. Anticipated positive emotions, social norms, environmental knowledge, and consequences of awareness all have a direct contribution to developing the intention to reduce waste. Waste reduction intentions, habits, and facilitating conditions are directly involved in developing waste reduction behaviors (reducing, reusing, and recycling). Our findings confirmed the importance of psychological and non-cognitive emotional factors in modifying the behaviors of young consumers to reduce, reuse, and recycle their FW to minimize its environmental, economic, and social consequences. The use of social media and marketing strategies will make it possible to change consumer behavior and reduce food waste. The results are essential for industry and scholars who want to promote waste reduction behavior by involving emotions via the media, as well as creating and modifying food waste habits in young consumers.

### 6.1. Implications

In this research, we undertook a thorough study on FWRB, applying the 3R waste reduction behavioral model along with an empirical examination of the cause of FWRB. This study primarily looked at the effects of emotional, social, cognitive, and psychological aspects on FWRB. Due to the serious consequences of FW on society and the environment, this study investigates the final link in the food waste supply chain, which is consumers. This study investigates some major gaps and exposes young consumers’ behaviors regarding food waste reduction. The following are the study’s primary theoretical implications:

#### 6.1.1. Theoretical Implications

The first gap identified was that most past research has concentrated on measuring the amount of food waste generated rather than investigating the causes of FWRB [10]. This study fills this gap in previous research on wasted food by observing the antecedents of the FWRB of young consumers. The second gap was that most previous research has concentrated on two components of FWRB, namely reuse and recycling. For instance, only a few studies have investigated people’s intentions to prevent food waste [90,91]. The present research took into account all three forms of FWRB by using the factors reducing, reusing, and recycling as endogenous components. Third, the current paper builds on the conceptual underpinnings of previous investigations on the motivators of FWRB. It was important because the majority of previous research had concentrated on a conceptual model, that is, the theory of planned behavior, which may not be suitable for reliably forecasting complicated human behavior [18,92,93] due to some missing aspects. Though the theory of planned behavior (TPB) is a prevalent model for predicting consumer intentions and behavior, it has been challenged in the past for failing to account for emotional and environmental components in FW reduction. In order to address TPB limitations, we employed the theory of interpersonal behavior to completely describe and anticipate consumer food waste behaviors by acknowledging crucial emotional, social, environmental, and external aspects as well as their reduction, reuse, and recycling behaviors [85]. The application of TIB and psychological aspects, for example, has resulted in theoretical advances in analyzing consumers’ intentions related to the consumer food waste reduction framework. The fourth is that, by analyzing the influential impact of emotions (anticipated positive emotions) in determining food waste reduction behavior, this study conceptually adds to the existing body of literature. The inclusion of anticipated positive emotions in the framework, in addition to cognitive components, thoroughly explains consumer food waste reduction behavior, addressing possible weaknesses in the predictive ability of TPB [18]. Researchers emphasizing food waste reduction as a responsible action in individuals’ minds may enhance an experience of pride in the case of food waste reduction. This emotion will set in motion a series of intentional activities that will motivate individuals to involve themselves in food waste reduction behaviors.

The fifth is that our observations also provide some insightful information regarding consumers’ food waste reduction behaviors. Food waste reduction behavior, for example, is a multidimensional concept in which an individual’s emotional condition, social assessment, cognitive awareness, and environmental attitude all play essential roles in shaping intentions toward sustainable FW reduction [13,94]. As well as the importance of cognitive components, such as emotions and habits, in explaining FW reduction, reuse, and recycling behavior, it was shown that combining habits and emotions might be a helpful technique for modifying behavior. Our data support the idea that increasing consumer participation in FW reduction, reuse, and recycling initiatives reduces its negative impact. The notion of reduce, reuse, and recycle has been shown to be effective in dealing with the complexity of food reduction behavior. The sixth is that our paper analyses environmental awareness as a precursor to food waste reduction behavior. We believe that being AOC will have a favorable link with intentions to reduce FW. Our data back the idea that consumers’ awareness of consequences produces emotions, as well as promotes socially desirable behaviors [95]. According to the findings, consumers tend to participate in food waste reduction activities when they are aware of the consequences of FW reduction. These outcomes are in line with past findings highlighting the consequences of food waste, such as economic hardship, unnecessary hunger, and environmental difficulties, all of which motivate consumers to act ethically and contribute to food reduction initiatives. The seventh is that our conclusions are grounded on data from young consumers, who have been the prime representative of the population and the most relevant samples for investigating pro-environmental FWRB. Because young consumers are the prospect, they might act as an indicator for the world’s people.

#### 6.1.2. Policy Implications

The study’s findings have far-reaching implications for practitioners and policymakers. The first is that the results of this investigation greatly aid in comprehending the relevance of the United Nations’ sustainable development goals (SDGs) [96]. The second is that the data shows that anticipating the positive emotions of consumers associated with food waste reduction might lead them to inculcate a deeper sense of responsibility. Resultantly, the incorporation of positive emotions becomes more beneficial, particularly in the case of socially required actions such as food waste reduction. Policymakers and practitioners, hence, need to emphasize the anticipated positive emotions such as a feeling of pride linked with food waste reduction in marketing tactics to urge consumers to intentionally participate in food waste reduction activities. Thirdly, consumer awareness of the consequences motivates them to reconsider their eating behaviors and their potential influence on society, the environment, and the economy. This may be accomplished by developing complete social awareness initiatives and employing social marketing strategies. Consumers can benefit from awareness campaigns that help them understand the significance of environmental issues such as FW and its negative impact on the economy and society. In particular, social media modes such as Facebook, Instagram, and YouTube can be employed to raise awareness of the consequences of FW and promote food waste reduction activities such as reducing, reusing, and recycling. It will also increase consumer environmental knowledge about reducing food waste due to economic losses, environmental harm, and social issues [96]. Fourthly, the findings emphasize the necessity of educating consumers regarding broad environmental concerns and precise environmental issues created by food waste behaviors.

Our study will also help policymakers build the awareness of young consumers related to food waste. Young customers can be educated on the necessity of engaging in sustainable FWRB by following the norms and regulations. As a result, efforts and marketing that place a strong emphasis on social influence have a greater chance of motivating customers to contribute to the reduction in food waste. Food waste behaviors should be addressed by educating consumers on the benefits of decreasing food waste. Fifth, practitioners should emphasize food security and imprint the repercussions of FW on the minds of young consumers to encourage them to participate in FWRB. Food waste reduction should be taught to the younger generation through social media, and they should be kept informed about the repercussions.

Furthermore, marketers, using current media such as social media will motivate young consumers to reduce FW and share it with others to raise their knowledge of the food waste problem. Sixth, practitioners at restaurants need to make approaches to reduce food waste through developing advertising messaging, marketing activities, and initiatives. To reduce waste due to larger orders than necessary, consumers must be informed of the pricing and amount of food that is adequate for their consumption and provides solutions for packing their uneaten meals to carry home and reuse again. Seventh, there are no specific legal and regulatory laws in Pakistan about food waste reduction management and methods of dealing with it. Eighth, promote food waste as financial waste. Ninth, encourage media movements highlighting the restaurant and hotel’s efforts to reduce FW to raise awareness, emotions, and shift habits.

### 6.2. Limitations and Future Recommendations

Like all research endeavors, this study also has certain limitations. First, the respondents were drawn from two cities throughout the country. Second, only the consumer viewpoint was examined; the opinions of restaurant/hotel personnel and management need to be studied to obtain an overall perspective on the issue. Third, we investigate the observed food waste reduction behavior in relation to whether or not consumers recollect their dining-out experiences. Fourth, in order to reduce food waste, we solely evaluated positive emotions and neglected the other parts of emotion. In the obtaining online wave and COVID situation, the present study was relevant to consumers who ate out, yet there are also household consumers who ordered meals online. Consumers at places such as workplaces, hospitals, and educational institutions might also be studied for future consideration. In addition, assessing income levels and urban–rural consumer comparisons may also give important information regarding the way food waste is regarded. The present research primarily focused on food waste, but further research might examine some other consumer waste causes, including disposable packaging and food expiry. Our study uses a quantitative technique to examine the elements that influence restaurant consumers’ food waste reduction, reuse, and recycling behaviors. Academics and practitioners could extend our paper by using our approach to examine other important issues, including food waste reduction [97,98], impacts of COVID-19 [99], carbon emissions [100], prediction of medicine prices [101], illegal waste disposal [102], and many others.

## Figures and Tables

**Figure 1 ijerph-19-06312-f001:**
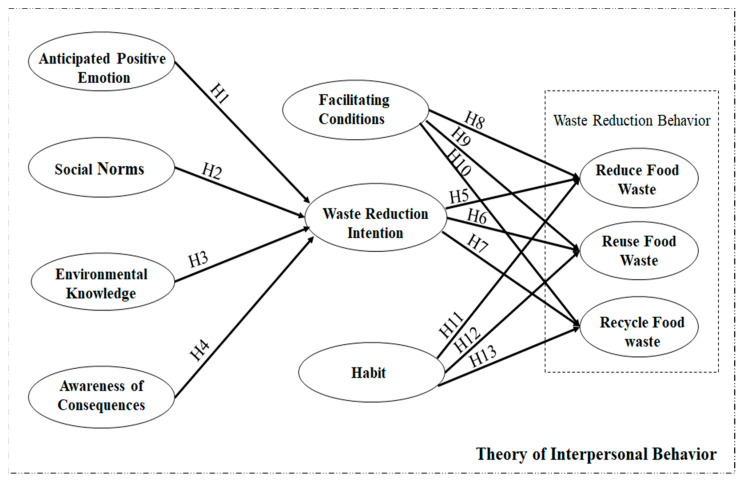
Conceptual Framework.

**Figure 2 ijerph-19-06312-f002:**
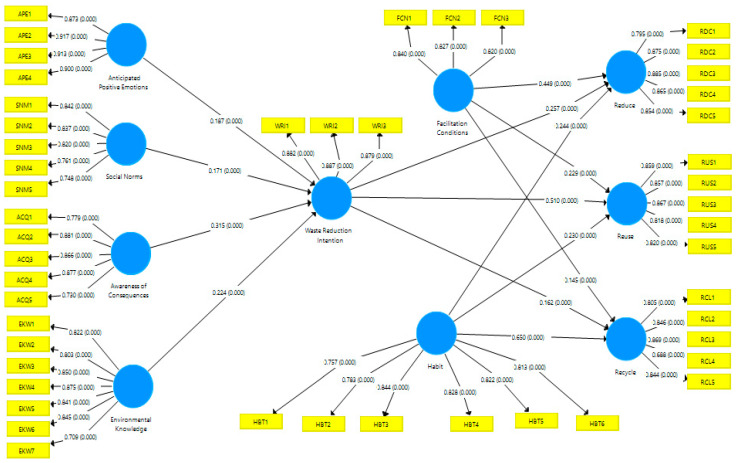
Structural Model.

**Table 1 ijerph-19-06312-t001:** Demographic statistics.

Demographic	Category	Percentage (Frequency)
Gender	Male	54% (572)
	Female	46% (491)
Age	<18 years	12% (129)
	18–25 years	42% (449)
	26–35 years	30% (313)
	36–45 years	10% (104)
	>45 years	6% (68)
Education	High school	12% (127)
	Professional degree/vocational school	7% (73)
	Bachelors	49% (522)
	Masters	28% (298)
	Doctorate	4% (43)

**Table 2 ijerph-19-06312-t002:** Measurement model results.

Constructs	Code	Outer Loadings	Cronbach’s Alpha	Composite Reliability	Average Variance Extracted
Threshold		>0.70	>0.70	>0.70	>0.50
Anticipated Positive Emotions	APE1	0.873	0.9226	0.9452	0.8117
	APE2	0.917			
	APE3	0.913			
	APE4	0.900			
Social Norms	SNM1	0.842	0.861	0.900	0.644
	SNM2	0.837			
	SNM3	0.820			
	SNM4	0.761			
	SNM5	0.748			
Environmental Knowledge	EKW1	0.822	0.919	0.936	0.676
	EKW2	0.803			
	EKW3	0.850			
	EKW4	0.875			
	EKW5	0.841			
	EKW6	0.845			
	EKW7	0.709			
Awareness of Consequences	AOC1	0.779	0.8844	0.162	0.6873
	AOC2	0.881			
	AOC3	0.866			
	AOC4	0.877			
	AOC5	0.730			
Facilitating Conditions	FCN1	0.840	0.775	0.868	0.687
	FCN2	0.827			
	FCN3	0.820			
Habits	HBT1	0.757	0.894	0.919	0.653
	HBT2	0.783			
	HBT3	0.844			
	HBT4	0.828			
	HBT5	0.822			
	HBT6	0.813			
Waste Reduction Intention	WRI1	0.882	0.858	0.913	0.779
	WRI2	0.887			
	WRI3	0.879			
Reduce Food Waste	RDC1	0.795	0.907	0.931	0.731
	RDC2	0.875			
	RDC3	0.885			
	RDC4	0.865			
	RDC5	0.854			
Reuse Food Waste	RUS1	0.859	0.899	0.925	0.713
	RUS2	0.857			
	RUS3	0.867			
	RUS4	0.818			
	RUS5	0.820			
Recycle Food Waste	RCL1	0.805	0.869	0.906	0.661
	RCL2	0.846			
	RCL3	0.869			
	RCL4	0.688			
	RCL5	0.844			

**Table 3 ijerph-19-06312-t003:** Discriminant Validity Analysis.

Constructs	1	2	3	4	5	6	7	8	9	10
1.Anticipated Positive Emotions	0.90									
2. Social Norms	0.72	0.80								
3. Environmental Knowledge	0.82	0.77	0.83							
4. Awareness of Consequences	0.71	0.76	0.75	0.82						
5. Facilitation Conditions	0.72	0.76	0.74	0.80	0.83					
6. Habit	0.70	0.64	0.70	0.65	0.68	0.81				
7. Waste Reduction Intention	0.73	0.74	0.78	0.73	0.68	0.60	0.88			
8. Reduce Food Waste	0.75	0.75	0.77	0.73	0.79	0.70	0.71	0.86		
9. Reuse Food Waste	0.75	0.75	0.79	0.74	0.73	0.69	0.80	0.76	0.84	
10. Recycle Food Waste	0.72	0.66	0.75	0.70	0.70	0.65	0.65	0.72	0.75	0.81

**Table 4 ijerph-19-06312-t004:** Hypotheses Testing.

Hypotheses	*β*	*t* Values	*p* Values
H1: Anticipated Positive Emotions → Waste Reduction Intention	0.187	4.576	0.000
H2: Social Norms → Waste Reduction Intention	0.171	4.345	0.000
H3: Environmental Knowledge → Waste Reduction Intention	0.224	6.600	0.000
H4: Awareness of Consequences → Waste Reduction Intention	0.315	7.095	0.000
H5: Waste Reduction Intention → Reduce Food Waste	0.257	8.715	0.000
H6: Waste Reduction Intention → Reuse Food Waste	0.510	18.827	0.000
H7: Waste Reduction Intention → Recycle Food Waste	0.162	6.147	0.000
H8: Facilitation Conditions → Reduce Food Waste	0.449	14.948	0.000
H9: Facilitation Conditions → Reuse Food Waste	0.229	8.337	0.000
H10: Facilitation Conditions → Recycle Food Waste	0.145	5.343	0.000
H11: Habit → Reduce Food Waste	0.244	8.440	0.000
H12: Habit → Reuse Food Waste	0.230	8.503	0.000
H13: Habit → Recycle Food Waste	0.650	26.558	0.000

**Table 5 ijerph-19-06312-t005:** Coefficient of determination (R^2^) and Q^2^.

Constructs	R^2^	Q^2^
Waste Reduction Intention	0.671	0.519
Reduce Food Waste	0.708	0.514
Reuse Food Waste	0.736	0.521
Recycle Food Waste	0.757	0.496

**Table 6 ijerph-19-06312-t006:** Effect size (f^2^).

Constructs	Waste Reduction Intention	Reduce	Reuse	Recycle
Anticipated Positive Emotions	0.033			
Social Norms	0.024			
Environmental Knowledge	0.054			
Awareness of Consequences	0.061			
Facilitation Conditions		0.292	0.084	0.037
Habit		0.103	0.101	0.875
Waste Reduction Intention		0.114	0.498	0.055

**Table 7 ijerph-19-06312-t007:** Indirect effect.

Hypotheses	*β*	*p* Values
H14: Anticipated Positive Emotions → Waste Reduction Intention → Waste Reduction Behavior	0.065	0.00
H15: Social Norms → Waste Reduction Intention → Waste Reduction Behavior	0.060	0.00
H16: Environmental Knowledge → Waste Reduction Intention → Waste Reduction Behavior	0.077	0.00
H17: Awareness of Consequences → Waste Reduction Intention → Waste Reduction Behavior	0.109	0.00

## Data Availability

Data are contained within the article.

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
