# Peer review of "Habit—Does It Matter? Bringing Habit and Emotion into the Development of Consumer’s Food Waste Reduction Behavior with the Lens of the Theory of Interpersonal Behavior"

_ijerph, 2022, doi:10.3390/ijerph19106312_

Round 1

Reviewer 1 Report

The paper has been improved but I consider that there are too many references and authors must reduce the number of them.

Reviewer 2 Report

Thanks to the authors of the manuscript and good luck in future research!

This manuscript is a resubmission of an earlier submission. The following is a list of the peer review reports and author responses from that submission.

Round 1

Reviewer 1 Report

In my opinion, there are too many references in the paper although it is not a bibligraphic review. It doesn't contribute to follow the paper.

Additionally, not many information about the survey is provided. For example, the questions and the possible answers. 

The survey participants are only University students who answered the survey online. This is not a representative sample. Authors should change the title and the aim of the paper in the abstract and introduction.l The paper must be shorter and better organized.

Author Response

response in attachment

Reviewer 2 Report

The study is interesting and concerns an important issue. The essence of the paper and its importance is to identify the elements that influence restaurant consumers' food waste reduction, reuse, and recycling behaviours. The authors correctly defined the aim of the study and formulate many research hypotheses. The research process is understandable and clearly described. The study collected 1063 responses, and PLS-SEM was used to analyse the hypotheses. A valuable element of the research is the wide literature review.  At the end of the study, the authors also indicated the implications, limitations and suggestions for further research. In my opinion, the reviewed paper is valuable in scientific and practical terms.

Author Response

response in attachment

Reviewer 3 Report

Research on how to reduce food waste is very important and necessary. It is also significant whether consumers are ready to reduce, reusing and recycling  the food  waste, and that affect it.

I wanted to recommend this as a Review article because there is very large theoretical part and very little research part.  But then I found your article in which a lot of text and the references are consistent with this article  - https://doi.org/10.3390/ijerph182312457. In the text changed only data and the consumers are mainly students from Pakistani universities. I understand that this has been a larger study, but then the results were also needed to be analysed within the overall study - how different aspects affect food waste reduce, reusing and recycling. But it can't be that this is already the third manuscript of one principle and text .

The guidelines for authors have not been taken into account in the manuscript  -  tables, figures and references. You are 176 references used - I think it's too much for this type of article.

Author Response

response in attachment
